**Data Availability Statement:** All relevant data with the exception of the personally identifiable information are deposited within public repository

# Epidemiology of dengue, chikungunya and Zika virus infections in travellers: A 16-year retrospective descriptive study at a tertiary care centre in Prague, Czech Republic

Milan Trojánek[1,2,3], Vyacheslav Grebenyuk[1,2,3]*, Zdenka Manďáková[4], Naděžda Sojková[5], Hana Zelená[6], Hana Roháčová[2], František Stejskal[1,2,7,8]

1 Department of Infectious Diseases, 2nd Faculty of Medicine, Charles University, Prague, Czech Republic, 2 Department of Infectious Diseases, University Hospital Bulovka, Prague, Czech Republic, 3 Department of Infectious Diseases and Travel Medicine, University Hospital Motol, Prague, Czech Republic, 4 Department of Epidemiology of Infectious Diseases, National Institute of Public Health, Prague, Czech Republic, 5 Department of Clinical Microbiology, University Hospital Bulovka, Prague, Czech Republic, 6 National Reference Laboratory for Arboviruses, Institute of Public Health in Ostrava, Czech Republic, 7 Institute of Immunology and Microbiology, 1st Faculty of Medicine, Charles University, Prague, Czech Republic, 8 Department of Infectious Diseases, Regional Hospital Liberec, Liberec, Czech Republic

* grebenyuk@seznam.cz

## Abstract

### Introduction

This study aims to describe the epidemiological characteristics of imported cases of dengue (DEN), chikungunya (CHIK), and Zika virus (ZIKV) infections in Czech travellers.

### Materials and methods

This single-centre descriptive study has retrospectively analysed data of patients with laboratory confirmed DEN, CHIK, and ZIKV infections diagnosed at the Department of Infectious, Parasitic and Tropical Diseases of the University Hospital Bulovka in Prague, Czech Republic from 2004 to 2019.

### Results

The study included a total of 313 patients with DEN, 30 with CHIK, and 19 with ZIKV infections. Most patients travelled as tourists:263 (84.0%), 28 (93.3%), and 17 (89.5%), respectively (p = 0.337). The median duration of stay was 20 (IQR 14–27), 21 (IQR 14–29), and 15 days (IQR 14–43), respectively (p = 0.935). Peaks of imported DEN and ZIKV infections were noted in 2016, and in 2019 in the case of CHIK infection. Most cases of DEN and CHIKV infections were acquired in Southeast Asia:212 (67.7%) and 15 (50%), respectively, while ZIKV infection was most commonly imported from the Caribbean (11; 57,9%).

Open Science Framework (DOI 10.17605/OSF.IO/GRW7Z. Available from: osf.io/grw7z).

**Funding:** The author(s) received no specific funding for this work.

**Competing interests:** The authors have declared that no competing interests exist.

## Conclusions

Arbovirus infections represent an increasingly significant cause of illness in Czech travellers. Comprehensive knowledge of the specific epidemiological profile of these diseases is an essential prerequisite for good travel medicine practice.

## Introduction

Over the past five decades, arbovirus infections have ceased to be sporadically occurring diseases becoming a global public health issue due to their expanding geographic distribution, increasing social and economic impacts on the population in the endemic regions, and importation risks associated with international travel and commerce, which may contribute to their further spreading beyond tropical areas Dengue fever is the most common arbovirus infection globally, causing over 390 million cases per year [1]. Chikungunya virus (CHIKV) was first described in 1952 and, in the following five decades, caused several minor outbreaks in Africa and Asia [2]. It reached global significance at the beginning of the 21st century, starting with the 2004 outbreak in Kenya and spreading to islands in the Indian Ocean, India, Southeast Asia, and eventually Latin America in 2013 [3]. Similarly, ZIKV, was first discovered in a Ugandan forest in 1947; it then re-emerged on the Yap Island in Micronesia in 2007, causing an extensive outbreak of dengue-like illness [4]. Subsequent major outbreaks occurred in French Polynesia in 2013 and in the Americas in 2015. It is estimated that in 2016 ZIKV caused a total of 7.6 million infections worldwide [5]. Based on the accumulating evidence pointing to an association of ZIKV infection with congenital defects and Guillain-Barré syndrome, on February 1, 2016, the World Health Organisation has declared the rapid spread of the virus a Public Health Emergency of International Concern [4].

In addition to causing substantial morbidity and socioeconomic disruption in the endemic regions, arbovirus infections pose a significant health risk to travellers visiting these regions for leisure, work, or other purposes. DEN infection remains the most important imported arbovirus disease in Czech travellers and its incidence is increasing, as demonstrated in a ten-year retrospective study previously conducted at the University Hospital Bulovka in Prague [6]. Another study conducted at the same site has shown that the dengue represents 8.2% of all travel-associated health issues, and that it is the most common cause of fever in Czech travellers returning from tropical destinations [7]. However, the literature on other imported arbovirus infections nationally, as well as reports from other Central and Eastern European countries remains scarce. This paucity of data has prompted the authors to conduct a retrospective analysis of DEN, CHIK and ZIKV infections to improve our current grasp of the epidemiological situation in the Czech Republic.

In addition to numerous travel-associated infections, autochthonous arbovirus transmission has been reported in parts of Europe and North America, where a competent mosquito vector is established [8–10]. Although the more efficient vector *Aedes aegypti* has not been introduced to most parts of the mainland Europe, the presence of *A. albopictus* has been documented in many countries, particularly along the Mediterranean basin [11]. According to entomologic surveillance studies from 2012 and 2016–2017, several mosquito larvae had been captured by ovitraps along main transit roads in the South Moravian, Central Bohemian, and Western Bohemian regions of the Czech Republic [12, 13]. Passive transportation from Southern European countries by motor vehicles seems to be a major risk factor as regards the introduction of these invasive mosquitoes to regions with a suitable environment. To date, no larvae or mosquitoes have been captured in human habitats, and there are no known cases of

autochthonous CHIK or ZIKV transmission in the Czech Republic. According to the modelling of the climate change impacts [14], CHIK transmission may become established in several regions of Central Europe in the next decades. Viraemic travellers could then become sources of local spreading of tropical arbovirus diseases in many parts of Europe.

As travellers from middle- to high-income countries generally have good access to resource-rich specialized medical care, they may serve as sentinels for tropical infections providing extensive data to improve our understanding of their rapidly changing epidemiology. The aim of this retrospective single-centre study was to describe epidemiological and clinical characteristics of imported dengue, chikungunya, and Zika virus infections diagnosed at the largest tertiary centre for travel medicine in the Czech Republic during the period from 2004 to 2019. The objectives were: to report all the imported cases of DEN, CHIK and ZIKV infections diagnosed at the Department of Infectious Diseases at the University Hospital Bulovka during the period from 2004 to 2019; to describe the epidemiological characteristics of the patients (reasons for travel, regions of acquisition, demographic characteristics) and the reported clinical features of arbovirus diseases; lastly, in the case of dengue fever, to analyse the associated serotypes.

## Materials and methods

This single-centre retrospective descriptive study included all patients with laboratory-confirmed DEN, CHIK, or ZIKV infections, who presented to the Department of Infectious, Parasitic and Tropical Diseases at the University Hospital Bulovka in Prague from January 2004 to December 2019. The department serves as a specialized tertiary care centre for tropical infections in Prague and the Central Bohemian Region, with a catchment area of 2.5 million population. The study was approved by the Ethical Committee of the University Hospital Bulovka (reference number 9214/EK-Z). Informed consent was obtained in all study participants. The clinical data were retrospectively extracted from the hospital's electronic medical records. The reported data included case incidence rates, monthly distribution, patient demographics (age and sex), travel characteristics (destination, duration, reason for travel, return date), symptoms and timing of illness, co-infections, clinical manifestations, and outcomes. Serotypes associated with selected dengue cases diagnosed from 2015 to 2017 were reported as well. DEN infection was classified according to the 2009 WHO Classification [15]. Chikungunya was classified within three categories: acute (symptom duration of ≤14 days), subacute (≤2 months), or chronic disease.

### Laboratory diagnosis

DENV infection was diagnosed by the detection of the NS1 antigen or viral RNA in acute serum samples or virus-specific IgM and IgG antibodies in acute and convalescent serum samples. CHIKV infection was diagnosed by the detection of viral RNA in serum, virus-specific IgM and subsequently IgG antibodies in acute and convalescent serum samples. Cases of ZIKV infection was diagnosed by the detection of viral RNA in blood, urine or sperm or virus-specific IgM and subsequently IgG antibodies and positivity of confirmatory VNT. Virological tests were performed at the Department of Clinical Microbiology of the University Hospital Bulovka in Prague. Virus-specific IgM and IgG antibodies were detected using commercially available ELISA kits: Dengue Fever Virus IgM Capture ELISA and Dengue indirect IgG ELISA (PanBio, Brisbane, Australia); NovaLisa Chikungunya Virus μ-capture ELISA and Chikungunya Virus IgG capture ELISA (NovaTec Immunodiagnostica, Germany), Anti-Zika IgM and IgG ELISA (Euroimmun, Germany). ZIKV VNT tests were performed at the National Reference Laboratory for Arboviruses (Institute of Public Health in Ostrava). The detection of

dengue NS1 antigen was performed using Platelia Dengue NS1 Antigen EIA test (BioRad Laboratories). Viral-specific RNA was detected using commercially available kits: Dengue Virus General-type real-time RT-PCR (Shanghai ZJ, Bio-Tech Co.), Chikungunya Virus Real Time RT-PCR (Shanghai ZJ Bio-Tech, China), and Zika virus Real-Time RT-PCR kit (Biolife).

### Statistical methods

Continuous variables are described by medians with interquartile ranges (IQR). Mann-Whitney or ANOVA tests were used for the comparison of continuous variables between two groups. Categorical variables are represented by absolute frequencies and proportions and analysed using Fisher's exact test. A p-value of $\leq 0.05$ was considered statistically significant. GraphPad PRISM 9.2.0 for Mac was used for all data analysis (GraphPad Software, San Diego California USA, www.graphpad.com).

## Results

### Disease incidence, trends, monthly distribution, and demographic characteristics of cases

The study included 313 patients with DEN, 30 with CHIK, and 19 with ZIKV infections. The median age was 34 years (IQR 28–41) in DEN patients, 34 (IQR 28–41) in CHIK patients, and 39 years (IQR 31–43) in ZIKV patients (p = 0.473). The male to female ratio was 1.19:1, 1.14:1, and 1.11:1 in the DEN, CHIK, and ZIKV subgroups, respectively (p = 0.986). The proportion of patients born in endemic countries was 12/313 (3.8%), 3/30 (10.0%), and 0 (0.0%), respectively (p = 0.118). The numbers of cases diagnosed per year are presented in Fig 1. The incidence of imported DEN infection increased significantly during the study period from 3 cases in 2004 to 42 cases in 2019 (p = 0.001). No statistically significant trends were observed for CHIK and ZIKV infections, as the incidence peaks associated with disease outbreaks in the visited areas. None of the patients reported previous vaccination against dengue virus or previous exposure to DEN, CHIK or ZIKV. Previous travel to dengue-endemic destinations was reported by 111/313 (35.5%) of DEN patients. The monthly distribution of imported DEN, CHIK and ZIKV infections in travellers is presented in Fig 2. There were no statistically significant differences in the monthly distribution of the diseases, with more cases generally diagnosed during the first and last months of the year (p = 0.185).

### Characteristics of travel and regions of acquisition

The reasons for travel are presented in Table 1. No statistically significant difference in the reasons for travel between DEN, CHIK, and ZIKV subgroups was identified in this study. The median duration of stay was 20 days (IQR 14–27) for patients with DEN infection, 21 days (IQR 14–29) for patients with CHIK infection, and 15 days (IQR 14–43) for patients with ZIKV infection, (p = 0.935). Regions and countries of acquisition are presented in Table 2.

### Epidemiological characteristics and diagnosis of dengue cases

Southeast Asia was the most common region of stay of the DEN patients. The most common destinations among business and VFR travellers were the Maldives (23/34, 67.6%) and Vietnam (10/14, 71.4%), respectively. Twenty-four cases of DEN infection were imported from the Maldives in 2012 and 2013, with the majority (22/24; 91.7%) acquired by construction site workers at a holiday resort on the Velaa island in Noonu atoll. Two cases of DEN infection were imported from Sub-Saharan Africa: one from Equatorial Guinea (2011) and one from Tanzania (2014).

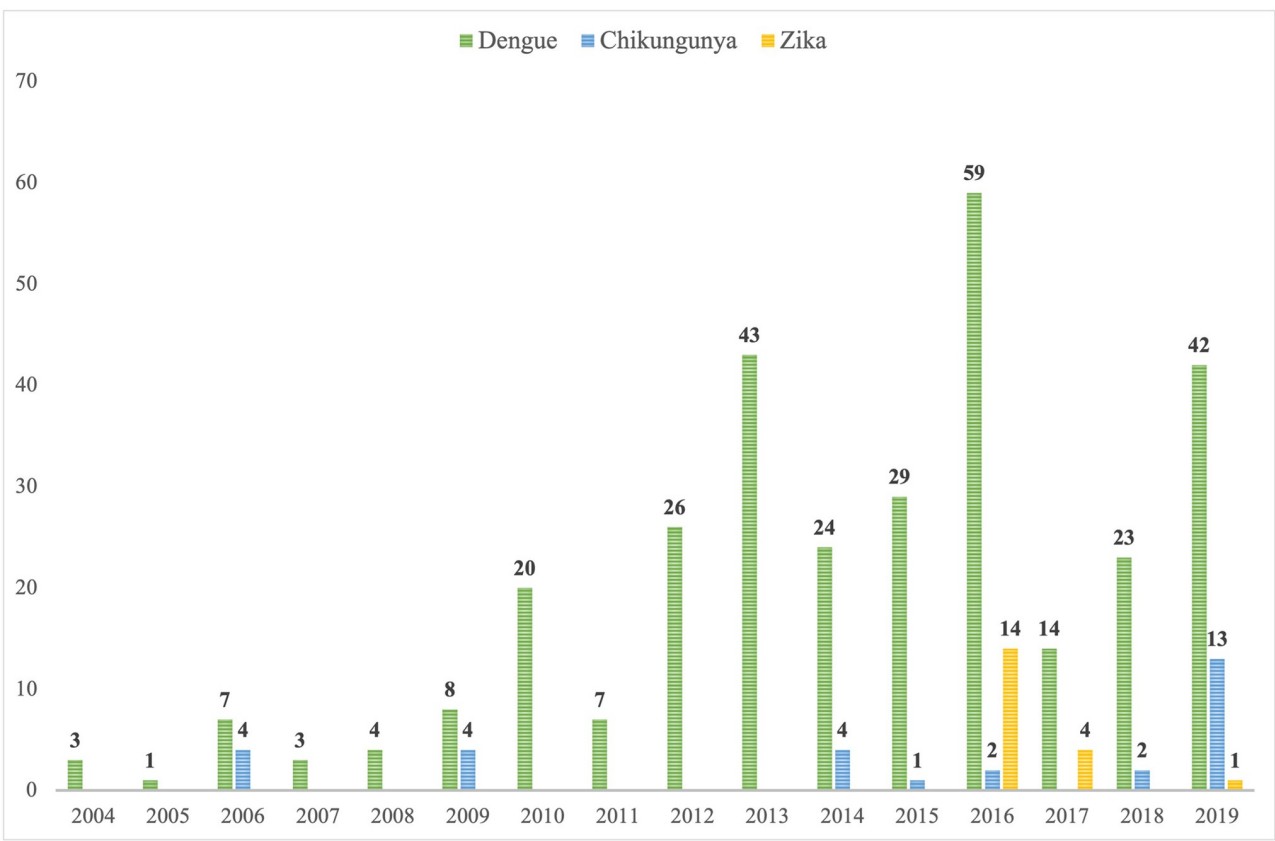

**Fig 1. Annual incidence rates of dengue, chikungunya and Zika cases.**

DENV-specific IgM and IgG antibodies were positive in 208/299 (69.6%) and 150/299 (50.2%) patients, respectively. The NS1 antigen was positive in 263/283 (92.9%) cases. Peripheral blood DEN RNA was detected in 130/166 (78.3%) patients. The diagnosis of acute dengue in IgG-positive cases was possible based on NS1 antigen detection in combination with PCR positivity in 57/150 cases (38.0%), NS1 positivity and IgM seroconversion in 58/150 (38.7%), NS1 alone in 6/150 (4.0%), and IgM seroconversion alone in 29/150 (19.3%). None of the patients reported previous infection with dengue or other tropical arboviruses. However, 38/150 (25.3%) of the IgG-positive patients had been vaccinated against tick-borne encephalitis (TBE) or yellow fever (YF). Dengue serotypes were available only in selected patients with a high viral load from 2015 to 2017 (Table 3).

### Epidemiological characteristics and diagnosis of chikungunya cases

The first CHIK cases were noted in February and April 2006 in 4 patients returning from Mauritius. In 2009 there were two cases imported from South Asia and two from Southeast Asia. A total of 7 cases were imported from the Americas in 2014 (Columbia, Guadeloupe), 2015 (Jamaica), 2016 (Mexico), and 2018 (Brazil). However, most CHIK cases (12/30; 40.0%) were imported from Thailand in 2019. Acute sera (<14 days from symptom onset) were available in 16/30 (53.3%) patients. CHIKV-specific IgM and IgG antibodies were detected in 9/16 (56.2%) and 4/16 (25.0%) of acute cases, respectively. Cases with negative acute serology were diagnosed by subsequent IgG seroconversion in the convalescent sera. In the remaining fourteen

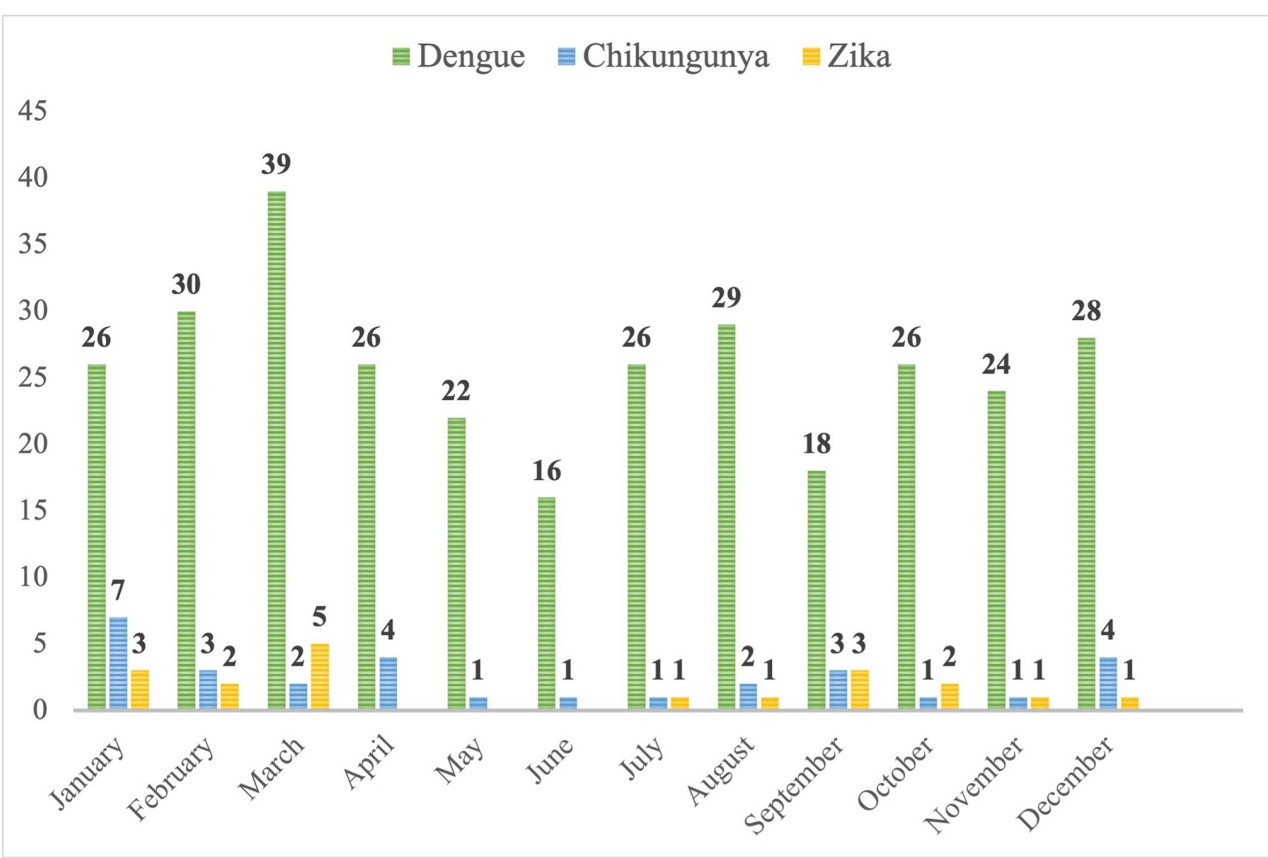

**Fig 2. January to December distribution of dengue, chikungunya and Zika cases.**

subacute cases IgM were positive in 12/14 (85.7%) and IgG in 13/14 (92.9%). Peripheral blood was tested for CHIK RNA in 3 patients and was positive in 2.

## Epidemiological characteristics and diagnosis of Zika cases

ZIKV infections were first noted in January 2016 in two patients returning from Martinique. Fourteen cases in total were diagnosed in 2016; all had been imported from the Caribbean or Central America. In the following year, three cases of ZIKV infection were diagnosed in patients returning from Cuba, and one case in a patient returning from the Maldives. The most recent case of Zika was diagnosed in November 2019 in a patient returning from Thailand. ZIKV-specific IgM and IgG antibodies were detected in 12/19 (63.1%) and 8/19 (42.1%) patients, respectively. VNT confirmation was performed in 16/19 (84.2%) cases with median

**Table 1. Reasons for travel in patients with acute DEN, CHIK, and ZIKV infections.**

|  | DENV infection | CHIKV infection | ZIKV infection |
|---|---|---|---|
| **Tourism** | 263 (84.0%) | 28 (93.3%) | 17 (89.5%) |
| **Business** | 34 (10.9) | 1 (3.3%) | 2 (10.5%) |
| **Visiting friends and relatives (VFR)** | 14 (4.5%) | 1 (3.3%) | 0 (0.0%) |
| **Other**[†] | 2† (0.6%) | 0 (0.0%) | 0 (0.0%) |

[†] Humanitarian aid (1), study reasons (1)

**Table 2. Regions and countries most frequently visited by patients with DEN, CHIK, and ZIKV infections.**

|  | DEN infection | CHIK infection | ZIKV infection | P-value |
|---|---|---|---|---|
| **Southeast Asia** | **212 (67.7%)** | **15 (50.0%)** | **1 (5.3%)** | **<0.001*** |
| Thailand | 90 (28.8%) | 12 (40.0%) | 1 (5.3%) | |
| Indonesia | 71 (22.7%) | 1 (3.3%) | 0 (0.0%) | |
| The Philippines | 18 (5.8%) | 0 (0.0%) | 0 (0.0%) | |
| Vietnam | 18 (5.8%) | 0 (0.0%) | 0 (0.0%) | |
| Cambodia | 8 (2.6%) | 0 (0.0%) | 0 (0.0%) | |
| Malaysia | 4 (1.3%) | 1 (3.3%) | 0 (0.0%) | |
| **South Asia** | **29 (9.3%)** | **3 (10.0%)** | **0 (0.0%)** | 0.597 |
| India | 18 (5.8%) | 3 (10.0%) | 0 (0.0%) | |
| Sri Lanka | 11 (3.5%) | 0 (0.0%) | 0 (0.0%) | |
| **Indian Ocean** | **42 (13.4%)** | **5 (16.7%)** | **1 (5.3%)** | 0.822 |
| The Maldives | 41 (13.1%) | 1 (3.3%) | 1 (5.3%) | |
| Mauritius | 0 (0.0%) | 4 (13.3%) | 0 (0.0%) | |
| **Central America** | **8 (2.6%)** | **1 (3.3%)** | **5 (26.3%)** | **0.001*** |
| Costa Rica | 5 (1.6%) | 0 (0.0%) | 1 (5.3%) | |
| Mexico | 3 (1.0%) | 1 (3.3%) | 2 (10.5%) | |
| Nicaragua | 0 (0.0%) | 0 (0.0%) | 2 (10.5%) | |
| **The Caribbean** | **11 (3.5%)** | **3 (10.0%)** | **12 (63.2%)** | **<0.001*** |
| Cuba | 7 (2.2%) | 0 (0.0%) | 3 (15.8%) | |
| Martinique | 0 (0.0%) | 0 (0.0%) | 7 (36.8%) | |
| Guadeloupe | 0 (0.0%) | 2 (6.7%) | 1 (5.3%) | |
| The Dominican Republic | 1 (0.3%) | 0 (0.0%) | 1 (5.3%) | |
| **South America** | **9 (2.9%)** | **3 (10.0%)** | **0 (0.0%)** | **0.033*** |
| Brazil | 6 (1.9%) | 2 (6.7%) | 0 (0.0%) | |
| Venezuela | 3 (1.0%) | 0 (0.0%) | 0 (0.0%) | |
| **Sub-Saharan Africa** | **2 (0.6%)** | **0 (0.0%)** | **0 (0.0%)** | 0.575 |
| Equatorial Guinea | 1 (0.3%) | 0 (0.0%) | 0 (0.0%) | |
| Tanzania | 1 (0.3%) | 0 (0.0%) | 0 (0.0%) | |

**Table 3. Dengue serotypes detected from 2015 to 2017.**

| Serotypes | Years | Countries | Number of cases | Total |
|---|---|---|---|---|
| DENV-1 | 2016 | Indonesia | 4 | 12 |
| | | Maldives | 4 | |
| | | Vietnam | 3 | |
| | | Brazil | 1 | |
| DENV-2 | 2016 | Indonesia | 6 | 13 |
| | | Thailand | 3 | |
| | | Cuba | 1 | |
| | 2017 | Sri Lanka | 2 | |
| | | Thailand | 1 | |
| DENV-3 | 2016 | Indonesia | 3 | 5 |
| | | Thailand | 1 | |
| | 2017 | Indonesia | 1 | |
| DENV-4 | 2015 | Sri Lanka | 1 | 2 |
| | 2016 | Indonesia | 1 | |

**Table 4. Clinical manifestations of dengue, chikungunya and Zika.**

|  | DEN | CHIK | ZIKV | P-value |
|---|---|---|---|---|
| Fever >38˚C | 311 (99.4%) | 30 (100.0%) | 8 (42.1%) | **<0.001**\* |
| Rash | 225 (71.9%) | 26 (86.7%) | 18 (94.7%) | **0.023**\* |
| Headache | 223 (71.2%) | 15 (50.0%) | 14 (73.7%) | **0.050**\* |
| Joint pain | 189 (60.4%) | 30 (100.0%) | 14 (73.7%) | **<0.001**\* |
| Joint swelling | 0 (0.0%) | 19 (63.3%) | 8 (42.1%) | **<0.001**\* |
| Muscle pain | 214 (68.4%) | 14 (46.7%) | 6 (31.6%) | **<0.001**\* |
| Abdominal pain | 36 (11.5%) | 1 (3.3%) | 1 (5.3%) | 0.137 |
| Diarrhoea | 91 (29.1%) | 7 (23.3%) | 8 (42.1%) | 0.363 |
| Vomiting | 49 (15.7%) | 2 (6.7%) | 0 (0.0%) | **0.027**\* |
| Pruritus | 137 (43.8%) | 7 (23.3%) | 6 (31.6%) | 0.064 |
| Dysgeusia | 57 (57.2%) | 2 (6.7%) | 0 (0.0%) | **0.011**\* |
| Mucosal bleeding | 49 (15.7%) | 0 (0.0%) | 0 (0.0%) | 0.001 |
| Dehydration | 25 (8.0%) | 2 (6.7%) | 0 (0.0%) | 0.557 |
| Conjunctivitis | 34 (10.9%) | 4 (13.3%) | 8 (42.1%) | **0.011**\* |
| Pharyngitis | 83 (26.5%) | 5 (16.7%) | 4 (21.1%) | 0.449 |
| Hepatomegaly | 26 (8.3%) | 2 (6.7%) | 1 (5.3%) | 0.781 |
| Splenomegaly | 12 (3.8%) | 0 (0.0%) | 0 (0.0%) | 0.382 |
| Lymphadenopathy | 43 (13.7%) | 5 (16.7%) | 6 (31.6%) | 0.102 |

titre of 1:32 (IQR 1:64 to 1:16). ZIKV RNA was detected by RT-PCR in various samples from 7 patients: blood in 2/13 (15.4%), urine in 6/15 (40.0%), and semen in 3/5 (60.0%). In two cases ZIKV-RNA was detected in sperm samples collected more than 60 days from symptom onset.

## Clinical features

The median period from symptoms onset to presentation was 4 days (IQR 3–6) in patients with DENV, 9 days (IQR 5–23) in patients with CHIK, and 7 days (IQR 3–19) in ZIKV infections) (p<0.001). The most common clinical manifestations of DEN, CHIK and ZIKV infections are presented in Table 4. The median duration of fever was five days (IQR 4–6) in dengue, four days (IQR 3–5) in chikungunya, and five days (IQR 4–6) in Zika virus infection (p = 0.035). Joint pain was reported by all patients with CHIK infection. Most commonly affected were small joints of hands or feet (26/30; 86.7%), ankles (12/30; 40.0%), and knees (12/30; 40.0%). Nineteen patients suffered chronic arthritis lasting over 2 months (63.3%). All Zika virus disease cases were symptomatic, and most reported either low-grade fever (8/19, 42.1%) or no fever (3/19 15.8%).

## Clinical course and outcomes

A total of 108/313 (34.5%) patients with DEN and 6/30 (20.0%) with CHIK infections were treated as inpatients (p = 0.154). All patients with ZIKV infection were treated as outpatients. Dengue warning signs were present in 73/313 (23.3%) of cases. One patient (0.3%) required intensive care for severe dengue with plasma leakage, hypotension, pleural effusion, and massive ascites. All CHIK and ZIKV infections presented with uncomplicated disease with no alarm signs. There were no lethal outcomes.

## Discussion

The presented study describes the epidemiological and clinical characteristics of patients with imported dengue, chikungunya, and Zika virus infections diagnosed in a tertiary care centre in

Prague, Czech Republic. With 362 patients, this constitutes one of the largest studies of imported arboviral infections in Central Europe. Substantial increments in the incidence of imported DENV, CHIKV, and ZIKV were observed from 2016 to 2019. These observations reflect the dramatic increase in the global burden of these diseases in recent years. An important contribution of this report is the direct comparison of epidemiological characteristics and clinical features of the three most common imported arboviral infections, which allows for pinpointing clinically relevant distinctions between the three otherwise very similar diseases.

## This study in the national and international contexts

During the study period, 566 cases of dengue fever were reported to the Czech national surveillance system of the National Institute of Public Health [16]. This study included 313 (55.3%) of these cases and thus provided a reasonable representation of the epidemiological situation in the whole country. As regards CHIK and ZIKV infections, the included patients comprise 85.7% and 100% of all nationally reported cases during 2004–2019, respectively. Such a high correspondence probably indicates missing cases on the national level, which may be related to low awareness of the disease among non-specialists and insufficient availability of diagnostic methods. The present study thus has the merit of shedding more light on the distribution of arboviral diseases imported from endemic areas to the Czech Republic, and to Europe more generally. According to the ECDC, the mean notification rate of chikungunya in the European Economic Area is <1 case per 1 million population, with the highest rates described in Belgium (5 per million) and Sweden (6 per million) [17]. In comparison, the nonfiction rate in the Czech Republic since 2014 has ranged from 0.1 to 1.4 cases per million population. However, there is a particular scarcity of data on tropical infections imported to Central and Eastern European countries. For instance, no cases of chikungunya have been reported in Poland or Slovakia as of 2019 [17, 18]. In Hungary, a country with a population size comparable to the Czech Republic, only 12 cases of chikungunya were described during the period from 2016 to 2019 [19].

## Incidence of imported DEN, CHIK and ZIKV infections and the observed trends

Imported dengue fever was diagnosed each year of the study period and there was a clear increasing trend during the later years. In contrast, the incidence of imported CHIK and ZIKV infection cases occurred in peaks following major epidemics in the tropical regions, and there were virtually no cases during other times. At the same time, there was also an apparent shift in the overall case burden to travellers returning from Southeast and South Asia during the most recent years. These changes are particularly important to European travellers, as they represent the largest and still growing proportion of international tourist arrivals to these regions [20].

## Epidemiological characteristics and clinical features of imported dengue cases

As shown in this study, dengue remains an important cause of morbidity in this population and its incidence continues to rise. The incidence peaks observed in 2016 and 2019 corresponded to increases in global dengue burden. During those years, higher transmission rates of the virus were reported both in Southeast Asia and the Americas [21, 22]. Incidence peaks were observed in mid-summer, late autumn, and early spring, probably reflecting increased rates of European outbound tourism during holiday seasons [23]. Apart from leisure tourism,

which represented the most common purpose of travel, a significant proportion of business travellers was noted. Twenty-two dengue cases were acquired by Czech construction workers during a single outbreak on the Velaa island in the Maldives [24]. DEN infections acquired in Sub-Saharan Africa represented only a negligible fraction of all dengue cases. DEN is known to circulate in this region [25] Still, the true incidence is notoriously hard to estimate due to underreporting and the high prevalence of cross-reactive immunity to other endemic flaviviruses that limit serological surveys. Nevertheless, several dengue outbreaks in African countries have been reported in recent decades [26]. DENV thus should still be considered in the differential diagnosis of fever in a traveller returning from Africa.

Clinically, dengue cases presented with acute febrile illness often accompanied by rash, headache, myalgias and arthralgias. However, skin involvement was less common than in chikungunya and Zika patients, and it was absent in 28% of dengue patients. Dengue warning signs were present in 23% of patients, and there was one case of severe dengue requiring intensive care.

## Serotypes associated with dengue cases imported from 2015 to 2017

The most common serotypes found in this study were DENV-1 and DENV-2. These findings are probably more reflective of the specific travelling patterns in Czech patients than the current epidemiological situation in the visited countries. Most Czech patients travel to Southeast Asia as tourists and typically visit coastal cities and the surrounding islands (e.g., Krabi, Bali, Ko Samui etc.) [6]. Due to the significant presence of the Vietnamese minority in the Czech Republic, there are many VFR travellers to Vietnam. These travellers are more likely to visit distant areas and rural regions. Interestingly, the only serotype isolated from travellers returning from Vietnam was DENV-1, which was the predominant serotype during the 2017 outbreak [27]. DENV-1 was also found in patients returning from the Maldives. The country experienced two large outbreaks of dengue in the recent decade. Only DENV-1 was detected in 2011, however, DENV-2 was reported in a Japanese couple during the 2018 outbreak [28, 29]. Confirmed previous infections with a different serotype or vaccination against DEN were not noted among patients in this study. The incidence of secondary dengue reported in previous studies in European travellers ranged from 17% to 43% [30]. Because of the retrospective nature of this study, some secondary dengue cases may have been missed. Previous exposure to other flaviviruses (including vaccination against TBE of YF) produces cross-reactive IgG, which further limits the distinction between primary and secondary cases. However, the absence of secondary cases may be partially explained by the fact that most patients were Czech nationals who reported no previous travel to dengue-endemic destinations. Furthermore, the live-attenuated tetravalent dengue vaccine has been available in the Czech Republic only since 2018, while it is still not licenced for use in travellers from non-endemic regions.

## Characteristics of imported chikungunya cases

Imported CHIK infection was first reported in the Czech Republic in travellers who had visited Mauritius during a large outbreak on the islands of the Indian Ocean in 2006 [31, 32]. The rates of imported chikungunya in the Czech Republic peaked in 2019, following another large-scale outbreak in Thailand with over 15,000 reported cases [33]. Twelve of the 13 cases reported in this study that date to 2019 had been acquired in Thailand. Southeast Asia has become the most common region of infection in European travellers due to its popularity among tourists and the high burden of CHIK in the area [17]. Acute chikungunya cases were characterized by fever, rash and arthralgias. CHIK cases tended to present relatively late (median 9 days from symptoms onset), presumably due to the relatively short duration of the

acute febrile phase in CHIK infection and the high incidence of subacute complications [34, 35]. Joint swelling and chronic arthritis developed in 63% of patients.

## Characteristics of imported Zika cases and implications for post-travel follow-up

Cases of imported ZIKV infection in Europe peaked in 2016, following a large outbreak in the Americas [8]. Since then, they have been declining at similarly rapid rates across all European countries. The dynamics observed in Czech travellers reflected the European trends. Notably, all Zika cases were symptomatic, typically presenting with low-grade fever and rash. Arthralgias were reported by 74% of patients, however, in contrast to chikungunya, joint swelling was significantly less common, and there was no progression to chronic arthritis. Among other distinctly common features were conjunctivitis, pharyngitis, lymphadenopathy, and diarrhoea.

Despite the overall decrease in ZIKV imported to Europe, the number of infections acquired in Southeast Asia has increased severalfold since 2016 [8]. Most of these cases were acquired in Thailand, the most popular tropical destination among Czech tourists. The rise in imported Zika cases may be partially accounted for by improved awareness of the disease and better availability of diagnostic methods since the association with congenital defects and Guillain-Barré syndrome was reported. Although the virus has been present in Southeast Asia for over 50 years, the true incidence of human infections is unknown. ZIKV is assumed to have primarily enzoonotic circulation in the region with only occasional spillover to humans [36]. No major outbreak has occurred to date, and only a few scattered cases have been reported in the literature [37]. The potential re-emergence of ZIKV in Southeast Asia has significant implications for travel medicine in Europe. Women of childbearing age and their partners travelling to the region should be counselled on the appropriate preventive measures, including mosquito bite precautions. Practitioners should also highlight the risks associated with pregnancy and sexual transmission. Notably, this study describes two cases of persisting ZIKV RNA positivity in sperm samples collected more than 60 days from symptom onset.

The limitations of this study are mainly attributable to its retrospective single-centre design. The study site serves primarily as a post-travel clinic for sick patients. Therefore, some mild or asymptomatic cases could have been missed. Previous exposure to tropical arboviruses in the included patients could not be reliably ruled out based on the available serological data. Finally, the findings of this study represent only the local epidemiological situation in the Czech Republic. However, they may shed more light on the distribution of arbovirus infections in the Central and Eastern European region, where a significant knowledge gap exists, and prompt more research on the topic in the neighbouring countries.

## Conclusions

Tropical arboviral infections are among the most important causes of travel-related morbidity. In addition to being the most popular tourist destination among Europeans, Southeast Asia has recently become the region with the highest risk of acquiring tropical arboviral infections. DENV infection represents a constant threat present during epidemics, as well as during the interim periods. Although travellers to Asia and Latin America are disproportionally affected, the risk of dengue is significant throughout the tropical zone, including Sub-Saharan Africa. Although CHIKV and ZIKV infections remain relatively rare causes of illness in Czech travellers, surges in imported cases invariably occur during major outbreaks in popular tourist destinations. Tourists may thus serve as sentinels for epidemiological surveillance of these diseases. With climate change and increase in global travel and trade, there is a potential for introduction of vector-borne pathogens to many parts of Europe, as well as future outbreaks in popular

tourist destinations all over the world. For this reason, physicians should become more familiar with the existing, as well as potential future arboviral threats.

## Acknowledgments

The authors would like to acknowledge Ms. Klara Hutkova, MA, who revised the manuscript for the English language. Our appreciation extends to all the staff of the Department of Infectious Diseases of the University Hospital Bulovka who participated in the management of included patients and provided primary data for this research through the hospital's medical records.

## Author Contributions

**Conceptualization:** Milan Trojánek, František Stejskal.

**Data curation:** Vyacheslav Grebenyuk, Zdenka Manďáková, Hana Zelená, František Stejskal.

**Investigation:** Milan Trojánek, Zdenka Manďáková, František Stejskal.

**Methodology:** Milan Trojánek, Naděžda Sojková, Hana Zelená, František Stejskal.

**Project administration:** Hana Roháčová, František Stejskal.

**Resources:** Naděžda Sojková.

**Supervision:** Hana Roháčová, František Stejskal.

**Visualization:** Vyacheslav Grebenyuk.

**Writing – original draft:** Vyacheslav Grebenyuk.

**Writing – review & editing:** Milan Trojánek, Vyacheslav Grebenyuk, Zdenka Manďáková, Naděžda Sojková, Hana Roháčová, František Stejskal.

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
