## [Decision Letter · Decision Letter 0]

20 Apr 2022

PONE-D-22-02970Epidemiology of dengue, chikungunya and Zika virus infection in travellers: a 16-year retrospective single-centre descriptive studyPLOS ONE

Dear Dr. Grebenyuk,

Thank you for submitting your manuscript to PLOS ONE. After careful consideration, we feel that it has merit but does not fully meet PLOS ONE’s publication criteria as it currently stands. Therefore, we invite you to submit a revised version of the manuscript that addresses the points raised during the review process.

As requested by the reviewer, try to provide a better clinical description of the various cases you count and in the same time please shorten as far as possible the description by providing synthetic tables as suggested by the reviewer.

We look forward to receiving your revised manuscript.

Kind regards,

Pierre Roques, Ph.D.

Academic Editor

PLOS ONE

Journal Requirements:

3. We note that you have stated that you will provide repository information for your data at acceptance. Should your manuscript be accepted for publication, we will hold it until you provide the relevant accession numbers or DOIs necessary to access your data. If you wish to make changes to your Data Availability statement, please describe these changes in your cover letter and we will update your Data Availability statement to reflect the information you provide

Additional Editor Comments (if provided):

Please indicated in the title that you refer to a Czech Republic institution

Reviewers' comments:

Reviewer's Responses to Questions

**Comments to the Author**

1. Is the manuscript technically sound, and do the data support the conclusions?

Reviewer #1: Yes

2. Has the statistical analysis been performed appropriately and rigorously? 

Reviewer #1: Yes

3. Have the authors made all data underlying the findings in their manuscript fully available?

Reviewer #1: No

4. Is the manuscript presented in an intelligible fashion and written in standard English?

Reviewer #1: Yes

5. Review Comments to the Author

Reviewer #1: In this study, authors describe the epidemiological characteristics of imported cases

of dengue (DENV), chikungunya (CHIKV), and Zika virus (ZIKV) infections in Czech travelers: 313 patients with DENV, 30 with CHIKV, and 19 with ZIKV infections. Most patients travelled as tourists – 263 (84.0%), 28 (93.3%), and 17 (89.5%), respectively. The median duration of stay was 20 (IQR 14-27), 21 (IQR 14-29), and 15 days (IQR 14-43), respectively.

The epidemiological and clinical characteristics described in the manuscript reflect the epidemiological situation of the countries visited and are not sufficiently precise. Important data are missing:

It would be interesting to describe more precisely the clinical manifestations of patients and to use the international classifications for dengue, chikungunya and Zika virus infection.

The breakdown between direct and serological diagnosis should be indicated for each arbovirus. For dengue, serological data should make it possible to differentiate primary dengue cases from others. It would also be interesting to know the dengue serotypes and compare them with the serotypes circulating in the countries visited at the time of travel.

In the end, Guadeloupe is not located in South America but in the Caribbean.

6. PLOS authors have the option to publish the peer review history of their article (what does this mean?). If published, this will include your full peer review and any attached files.

Reviewer #1: No

---

## [Author Response · Author response to Decision Letter 0]

4 Jun 2022

Editor

As requested by the reviewer, try to provide a better clinical description of the various cases you count and in the same time please shorten as far as possible the description by providing synthetic tables as suggested by the reviewer.

Accepted. We have provided detailed clinical description of the cases and shortened the manuscript as far as possible without leaving out the essential parts. 

Editor

Please indicated in the title that you refer to a Czech Republic institution.

Accepted, the title has been changed.

Journal Requirements

Please ensure that you have an ORCID iD and that it is validated in Editorial Manager. 

ORCID iD added and validated in Editorial Manager. 

In your Data Availability statement, you have not specified where the minimal data set underlying the results described in your manuscript can be found. PLOS defines a study's minimal data set as the underlying data used to reach the conclusions drawn in the manuscript and any additional data required to replicate the reported study findings in their entirety. All PLOS journals require that the minimal data set be made fully available.

The study’s minimal underlying data set has been made available within the public repository Open Science Framework (osf.io/grw7z DOI 10.17605/OSF.IO/GRW7Z).

Reviewer #1

#1 It would be interesting to describe more precisely the clinical manifestations of patients and to use the international classifications for dengue, chikungunya and Zika virus infection.

Accepted. A detailed description of the clinical manifestations is provided, including the international classification for dengue cases (i.e., classic dengue, dengue with warning signs, severe dengue) as per the 2009 WHO classification. Based on symptom duration, chikungunya cases were classified into acute, subacute, and chronic diseases. In the revised manuscript, we have specified that all chikungunya and Zika cases were symptomatic and clinically uncomplicated. We are not aware of any other stable international classification of chikungunya or Zika. The eleventh edition of the International Classification of Diseases contains one designation for Chikungunya virus disease and one for Zika virus disease.

#2 The breakdown between direct and serological diagnosis should be indicated for each arbovirus. 

Accepted. We have specified the diagnostic modalities in the text.

#3 For dengue, serological data should make it possible to differentiate primary dengue cases from others. 

Accepted, serological data were added to the text. Unfortunately, it is not always possible to differentiate primary and secondary dengue cases as a substantial proportion of Czech travellers is vaccinated against other arboviral diseases (including tick-borne encephalitis and yellow fever), which may lead to cross-reactive IgG positivity. Therefore, we have provided the data on the vaccination status in the revised manuscript. However, due to the study’s retrospective design, precise data on vaccination status or previous arbovirus exposure were not available. We have clarified this point in the discussion. 

#4 It would also be interesting to know the dengue serotypes and compare them with the serotypes circulating in the countries visited at the time of travel.

We accept this point and agree that it would add much value to the manuscript. Unfortunately, we do not routinely perform dengue serotyping at our centre due to the limited resources. Nevertheless, there was an effort to establish serotype differentiation in selected dengue patients (i.e., those with PCR positivity and sufficient viral load) from 2015 to 2017. We have added the available data to the manuscript. A comparison with the serotypes circulating at the time is provided in the discussion.

#5 In the end, Guadeloupe is not located in South America but in the Caribbean.

Accepted.

---

## [Decision Letter · Decision Letter 1]

17 Aug 2022

PONE-D-22-02970R1Epidemiology of dengue, chikungunya and Zika virus infection in travellers: a 16-year retrospective descriptive study at a tertiary care centre in Prague, Czech RepublicPLOS ONE

Dear Dr. Grebenyuk,

Thank you for submitting your manuscript to PLOS ONE. After careful consideration, we feel that it has merit but does not fully meet PLOS ONE’s publication criteria as it currently stands. Therefore, we invite you to submit a revised version of the manuscript that addresses the points raised during the review process.

The interest of such a study as outlined by the reviewer 2 and remembered by the reviewer 3 deserved to be better explained in the introduction of this article to answer to the very well and sharp comment of the reviewer 3. Please correct some of the mistake indicated by the reviewer 2 to correctly described al the figures or table included in the article within the result text. IN addition take care to answer point by point to the reviewer 2 and to correctly report the limitation of the retrospective study (not international but national with a potential impact on international reader).

We look forward to receiving your revised manuscript.

Kind regards,

Pierre Roques, Ph.D.

Academic Editor

PLOS ONE

Reviewers' comments:

Reviewer's Responses to Questions

**Comments to the Author**

1. If the authors have adequately addressed your comments raised in a previous round of review and you feel that this manuscript is now acceptable for publication, you may indicate that here to bypass the “Comments to the Author” section, enter your conflict of interest statement in the “Confidential to Editor” section, and submit your "Accept" recommendation.

Reviewer #2: All comments have been addressed

Reviewer #3: (No Response)

2. Is the manuscript technically sound, and do the data support the conclusions?

Reviewer #2: Yes

Reviewer #3: Partly

3. Has the statistical analysis been performed appropriately and rigorously? 

Reviewer #2: Yes

Reviewer #3: N/A

4. Have the authors made all data underlying the findings in their manuscript fully available?

Reviewer #2: Yes

Reviewer #3: No

5. Is the manuscript presented in an intelligible fashion and written in standard English?

Reviewer #2: Yes

Reviewer #3: No

6. Review Comments to the Author

Reviewer #2: I. Personalized summary of the research work carried out

This is a 16-year prospective study from 2004 to 2019, which took place in a tertiary care center in Prague, Czech Republic. As part of this work, during the relevant study period, the authors propose:

In a first step, to notify the annual number of cases of imported and diagnosed infections, as well as their monthly distribution over the twelve months of the year. These infections concern the three arboviruses dengue, chikungunya and Zika;

In a second step, to define some epidemiological characteristics and clinical features (reasons for travel, region or localities visited, associated clinical manifestations) of patients infected with these three arboviruses already mentioned.

Finally, in the case of dengue fever, to determine the associated serotypes from 2015 to 2017.

II. Summary of the interest of this research article

This study relating to the epidemiological description of cases of infections by dengue, chikungunya and Zika viruses imported to Prague in the Czech Republic through travelers, finds its interest and its relevance, in the context of globalization and the search for solutions at the international level to overcome emerging or re-emerging infectious and vector-borne diseases. These diseases, whether associated or not, are responsible for a heavy burden in terms of morbidity in the world in general and in tropical countries with limited resources in particular.

This work has the merit of giving some visibility on the distribution of infections due to arboviruses and imported from endemic tropical areas to Europe more generally and to the Czech Republic more particularly due to population movements via travel. Indeed, these imported infections such as Dengue, Chikungunya and Zika, which constitute tropical diseases with epidemic potential, require monitoring and scientific research not only at the national level in endemic countries but also internationally in view of their eradication.

In this work, the authors document the regions and countries most visited by patients with DENV, CHIKV, and ZIKV infections, which gives an overview of the impact of these contaminations in terms of public health on a scale of the Czech Republic but more generally suggests the significant impact on a global scale due to travel resulting in significant population movements.

In addition, the results of this work show the need for effective surveillance of populations returning from travel to prevent the spread of these infections in Europe due to the existence of a potentially competent transmission vector.

In countries with limited resources, the clinical manifestations associated with dengue, for example, have similarities to those due to malaria. Also, due to limited resources, in these countries patients are most often treated on the basis of clinical signs alone. Also, the differential diagnosis of infections due to these different arboviruses is not carried out and many of these tropical diseases such as dengue fever are classified in the group of neglected tropical diseases. This present work has the merit of carrying out laboratory research and diagnosis. This makes it possible to give the true incidence of these different vector-borne infectious diseases and to trace their possible international spread through travellers.

III. Problems and comments raised

Some comments have already been taken into account by the previous reviewer, however the following points should be noted.

1#. The case of infection or re-infection or vaccination of the patients in the study are not documented, certainly because of the retrospective nature of the study, which limits the scientific exploitation of the results of this work.

2#. The same is true of the Caucasian or non-Caucasian nature of the study population. Indeed, this information could make it possible to verify whether or not there is a link between the severity of the clinical manifestations and the natural history of the disease in the two study groups.

3#. In the context of this study, patients with, among other things, other imported infections were excluded from the study. However, taking this information into account would have added to the knowledge of co-infections such as malaria-dengue fever and zika-dengue fever, sometimes noted in the literature. This could have documented the correlation between the clinical signs and the share of each of these tropical infections according to the geographical origin of the travel of the infected patients. Something that is often difficult to do in endemic areas due to the lack of laboratory diagnostic means.

4#. The results of figure 1 do not appear explicitly in the results, not even those mentioned in particular in the discussion (line 226-228 in other).

5#. The title of figure 4: "Seasonality of imported arboviral" lacks precision. The seasons are known as for example summer, spring, autumn and winter. In fact, the title could be "January to December distribution of imported arboviral".

IV. Conclusion

Despite the limitations due to insufficiently documented information, certainly linked to the retrospective nature of the study, this work, which meets the publication criteria required by PLOS One, also presents a notorious interest already indicated, for which it deserves to be accepted for publication.

Reviewer #3: This manuscript describes the epidemiological characteristics of imported human arbovirus infections (dengue, chikungunya and Zika virus) in a tertiary care centre in Prague, Czech Republic in 2004 to 2019. There is strength in the work, with particularly contribution evident in providing an indication of the situation of arbovirus infections in Prague. However, the work does not accurately reflect the epidemiological situation of the infections neither on local (Prague) nor national (Czech Republic) level. The author did not mention whether these records represent all the arbovirus notifications reported in the Prague/ Czech. So, the scope of the work deals with ‘local’ topic, rather than genuinely ‘international’ one. I encourage authors to publish this paper as a national report or in a national journal after improving the manuscript.

The topic is interesting, and there are the foundations for a useful study. However, there is a lack of proper structure (especially in background and discussion) and detail about the knowledge gap, aims and objectives the methodology (collecting and recording the notifications). The analysis of data is also very much underdone (i.e. overly simple analysis has been performed). Further, the manuscript needs extensive revision for language and grammar

to improve the readability.

The author did not discuss the risk of local transmission of these infections in Czech Republic by way of infected travellers and evaluate the other risk factors that might coincide in time or space to trigger a local outbreak. The risk of local transmission is always more critical the than the importation risks of arboviruses. A better contextualise the work in terms of public health risk would greatly improve the value of the work. Hopefully the author can remedy these points and improve the manuscript.

7. PLOS authors have the option to publish the peer review history of their article (what does this mean?). If published, this will include your full peer review and any attached files.

Reviewer #2: No

Reviewer #3: No

---

## [Author Response · Author response to Decision Letter 1]

12 Oct 2022

Reviewer Comments:

Reviewer #2

1#. The case of infection or re-infection or vaccination of the patients in the study are not documented, certainly because of the retrospective nature of the study, which limits the scientific exploitation of the results of this work.

Accepted. We have reviewed the medical records of all the dengue cases included in the study and did not find any patient, who reported previous infection with dengue or other tropical arbovirus. Obviously, some cases might have been missed as clinicians do not always explicitly ask the patients about previous infections. However, the vast majority of patients were short-term tourists of Czech nationality. Secondary infections in similar populations are relatively uncommon. In addition, only 111 patients with dengue reported previous stay in an endemic region. Although a tetravalent dengue vaccine (Dengvaxia) has been licensed in many countries, it has not been registered in the Czech Republic up until December 2018 and is still not recommended for use in travellers. 

2#. The same is true of the Caucasian or non-Caucasian nature of the study population. Indeed, this information could make it possible to verify whether or not there is a link between the severity of the clinical manifestations and the natural history of the disease in the two study groups. 

Accepted, data on the origin of the study participants have been added to the manuscript. No significant differences in the severity of clinical manifestations or natural history of the disease were found. However, the study may be underpowered to show an association as only a minority of patients were non-Caucasians.

3#. In the context of this study, patients with, among other things, other imported infections were excluded from the study. However, taking this information into account would have added to the knowledge of co-infections such as malaria-dengue fever and zika-dengue fever, sometimes noted in the literature. This could have documented the correlation between the clinical signs and the share of each of these tropical infections according to the geographical origin of the travel of the infected patients. Something that is often difficult to do in endemic areas due to the lack of laboratory diagnostic means.

We understand this point and agree with the Reviewer, that it would be interesting and informative to report co-infections with two or more different imported pathogens. However, as we noted in the previous report from our centre (Grebenyuk V, et al. Epidemiol Mikrobiol Imunol. 2021), endemic tropical infections are responsible for a relatively small fraction of the overall morbidity in Czech travellers returning from tropical destinations. Particularly chikungunya and Zika represent rare infections comprising <1% of all travel-associated health problems. Considering this, the likelihood of a co-infection with another endemic pathogen should be very low. Moreover, Zika serology has become available in our centre only since 2016, although the disease has been known to cause outbreaks for almost a decade before that. The rates of imported dengue and malaria are considerably higher making co-infection by these two pathogens more likely. However, a 2018 systematic review (Salam N, et al. BMC Public Health, 2018) found only 2 cases diagnosed in Europe (Spain and France). After carefully reviewing all cases DEN, CHIK and ZIKV infections diagnosed at our centre during the study period we found no co-infections. The Methodology has been revised accordingly.

4#. The results of figure 1 do not appear explicitly in the results, not even those mentioned in particular in the discussion (line 226-228 in other).

Accepted, these results are now mentioned in the text.

5#. The title of figure 4: "Seasonality of imported arboviral" lacks precision. The seasons are known as for example summer, spring, autumn, and winter. In fact, the title could be "January to December distribution of imported arboviral".

Accepted, the title has been changed as suggested.

Reviewer #3: 

This manuscript describes the epidemiological characteristics of imported human arbovirus infections (dengue, chikungunya and Zika virus) in a tertiary care centre in Prague, Czech Republic in 2004 to 2019. There is strength in the work, with particularly contribution evident in providing an indication of the situation of arbovirus infections in Prague. However, the work does not accurately reflect the epidemiological situation of the infections neither on local (Prague) nor national (Czech Republic) level. The author did not mention whether these records represent all the arbovirus notifications reported in the Prague/ Czech. So, the scope of the work deals with ‘local’ topic, rather than genuinely ‘international’ one. I encourage authors to publish this paper as a national report or in a national journal after improving the manuscript.

The topic is interesting, and there are the foundations for a useful study. However, there is a lack of proper structure (especially in background and discussion) and detail about the knowledge gap, aims and objectives the methodology (collecting and recording the notifications). The analysis of data is also very much underdone (i.e. overly simple analysis has been performed). Further, the manuscript needs extensive revision for language and grammar

to improve the readability.

The author did not discuss the risk of local transmission of these infections in Czech Republic by way of infected travellers and evaluate the other risk factors that might coincide in time or space to trigger a local outbreak. The risk of local transmission is always more critical the than the importation risks of arboviruses. A better contextualise the work in terms of public health risk would greatly improve the value of the work. Hopefully the author can remedy these points and improve the manuscript.

We appreciate the feedback provided by the Reviewer and are grateful for the valuable points raised. The single-centre nature of the study is a significant limitation, and the presented data do not represent all cases imported to the Czech Republic. To compensate for this shortcoming, we have requested the National Institute of Public Health to provide complete data on arbovirus epidemiology. These data have been added to the Discussion. 

The Healthcare system in the Czech Republic is highly centralized, and many rare diseases tend to be concentrated in a few large institutions. The Department of Infectious Diseases of the University Hospital Bulovka is the principal referral centre for infectious diseases not only for Prague and the Central Bohemian region but, effectively, for the whole country. Although this work does not represent all the infections notified in the Czech Republic, it does include the vast majority of them (especially in the case of chikungunya and Zika). Thus, we believe our data may reflect the epidemiological situation in the country. Furthermore, since there is a scarcity of data on tropical arboviral infections imported to Central and Eastern Europe, the authors maintain that the study provides valuable insight into the arbovirus epidemiology in the whole region and may prompt more research on the topic in the neighbouring countries. 

Based on the Reviewer’s suggestions, we have made some changes to the manuscript. The risk of local transmission and the introduction of the mosquito vectors were added to the Introduction. The paragraph discussing the risk of sexual transmission of ZIKV has been revised in the Discussion. Also, the aim, objectives and knowledge gap have been described in more detail in the Introduction and Discussion. The methodology was likewise revised following the provided comments. Results and Discussion were divided into subsections to improve readability. Finally, the manuscript has been revised for the English Language by a post-doc in English Language and Linguistics.

---

## [Decision Letter · Decision Letter 2]

27 Jan 2023

Epidemiology of dengue, chikungunya and Zika virus infections in travellers: a 16-year retrospective descriptive study at a tertiary care centre in Prague, Czech Republic

PONE-D-22-02970R2

Dear Dr. Grebenyuk,

We’re pleased to inform you that your manuscript has been judged scientifically suitable for publication and will be formally accepted for publication once it meets all outstanding technical requirements.

Kind regards,

Pierre Roques, Ph.D.

Academic Editor

PLOS ONE

Additional Editor Comments (optional):

Sorry for the delay

Reviewers' comments:

Reviewer's Responses to Questions

**Comments to the Author**

1. If the authors have adequately addressed your comments raised in a previous round of review and you feel that this manuscript is now acceptable for publication, you may indicate that here to bypass the “Comments to the Author” section, enter your conflict of interest statement in the “Confidential to Editor” section, and submit your "Accept" recommendation.

Reviewer #2: All comments have been addressed

2. Is the manuscript technically sound, and do the data support the conclusions?

Reviewer #2: Yes

3. Has the statistical analysis been performed appropriately and rigorously? 

Reviewer #2: Yes

4. Have the authors made all data underlying the findings in their manuscript fully available?

Reviewer #2: Yes

5. Is the manuscript presented in an intelligible fashion and written in standard English?

Reviewer #2: Yes

6. Review Comments to the Author

Reviewer #2: Tropical arboviruses constitute an important cause of potentially serious microbial infections not only in indigenous populations of the tropics but also in travelers who stay in these tropical regions, in particular the Cheques.

This work presents the interest of describing the epidemiological monitoring of these infections in these travellers. This information is important in the but to provide reliable and regular data allow in a "One Health" context not only to measure in view of controlling the introduction of pathogenic agents with vector transmission in many parts of Europe but also to raise the alarm about the occurrence of future epidemics in popular tourist destinations around the world that do not always have effective surveillance systems.

7. PLOS authors have the option to publish the peer review history of their article (what does this mean?). If published, this will include your full peer review and any attached files.

Reviewer #2: No

---

## [Editor Report · Acceptance letter]

10 Feb 2023

PONE-D-22-02970R2 

Epidemiology of dengue, chikungunya and Zika virus infections in travellers: a 16-year retrospective descriptive study at a tertiary care centre in Prague, Czech Republic 

Dear Dr. Grebenyuk:

I'm pleased to inform you that your manuscript has been deemed suitable for publication in PLOS ONE. Congratulations! Your manuscript is now with our production department. 

Kind regards, 

on behalf of

Dr. Pierre Roques 

Academic Editor

PLOS ONE